# Uncovering Sets of Maximum Dissimilarity on Random Process Data

**Miguel de Carvalho**                                             *miguel.decarvalho@ed.ac.uk*
*School of Mathematics, University of Edinburgh, UK*
*CIDMA, Universidade de Aveiro, Portugal*

**Gabriel Martos**                                                           *gmartos@utdt.edu*
*Departamento de Matemática y Estadística, Universidad Torcuato Di Tella, Argentina*

**Reviewed on OpenReview:** *https://openreview.net/forum?id=ntWCJrlDD8*

## Abstract

The comparison of local characteristics of two random processes can shed light on periods of time or space at which the processes differ the most. This paper proposes a method that learns about regions with a certain volume, where the marginal attributes of two processes are less similar. The proposed methods are devised in full generality for the setting where the data of interest are themselves stochastic processes, and thus the proposed method can be used for pointing out the regions of maximum dissimilarity with a certain volume, in the contexts of point processes, functional data, and time series. The parameter functions underlying both stochastic processes of interest are modeled via a basis representation, and Bayesian inference is conducted via an integrated nested Laplace approximation. The numerical studies validate the proposed methods, and we showcase their application with case studies on criminology, finance, and medicine.

## 1 Introduction

Everyday millions of data patterns flow around the world at unprecedented speed, thus leading to an explosion on the demand for modeling stochastic process data—such as point processes, time series, and functional data; each of these types of data plays a key role in machine learning, as can be seen, for instance, from the recent papers of Berrendero et al. (2020), Faouzi & Janati (2020), and Xu et al. (2020). Hand in hand with this shock on demand arrived a pressing need for the development of data-intensive methods, techniques, and algorithms for learning and comparing random processes.

**Problem**. We deal with the following general problem on the comparison of stochastic processes:

> *For a pair of random processes, what is the region—with a given volume—where they statistically differ the most?*

Since the target of interest consists of a set that fulfils an optimization criterion—i.e. a period of time or region over which a marginal feature of two stochastic processes differ the most—some of the main concepts in this paper can be framed as an optimization problem over a set function. Unlike standard functions that assign a number to another number, set functions assign a value to each set. For example, a probability measure $F(A) = \Pr(A)$, where $A$ is an event (say, $A = [0, \infty)$), calculates the probability of that event; another example is the counting measure $F(A) = \#A$, which simply counts the number of elements in a set (for instance, $A = \{1, 2, 3\}$). As these examples illustrate, the sets of interest can be either discrete or defined on a continuum. For the purposes of this paper, since the goal is to identify a region with a given volume, we will focus on sets defined on a continuum.

The canonical problem in set function optimization is of the type,

$$
\begin{aligned}
\max_{A \subseteq \mathscr{A}} \quad & F(A) \\
\text{s.t.} \quad & A \in \mathscr{F},
\end{aligned}
\tag{1}
$$

where $F : \mathcal{P}(\mathscr{A}) \to \mathbf{R}$ is a set function, $\mathscr{A}$ is a set, $\mathcal{P}(\mathscr{A})$ denotes the powerset of set $\mathscr{A}$, and $\mathscr{F} \subseteq \mathcal{P}(\mathscr{A})$ is the collection of feasible sets defining the constraint. Optimization problems over set functions—such as (1)—are commonplace in machine learning (e.g., Krause, 2010). For a recent review on the theory of discrete set function optimization see Wu et al. (2019). Most state of the art developments have been made on *discrete* or *combinatorial set function optimization*, especially on the class of submodular functions (e.g., Goldengorin, 2009). As a byproduct, our paper provides one of the first steps towards *continuous set function optimization* as here the aim will be to solve (1) when $\mathscr{F}$ is a family of Borel subsets of a compact set $T \subset \mathbf{R}^d$.

As it will be seen in Section 2, the objective set function of interest in our case will be a measure of proximity between marginal features of the pair of stochastic processes of interest, whereas the collection of feasible sets introduces the constraint on the 'size' of the feasible regions—i.e. periods of time or space—over which the comparison is made.

**Main contributions**:

1. We pioneer the study, formulation, and analysis of the learning problem of tracking down sets of maximum discrimination as described above—and formally defined in Sections 2–3.

2. In its most standard version, the proposed learning problem is shown to be equivalent to a continuous set function optimization on a monotone modular function, under a Lebesgue measure constraint. Hence, as a byproduct, the paper contributes to the literature on set function optimization which is mostly focused on a discrete and combinatorial framework, with a particular emphasis on monotone submodular functions (e.g., Nemhauser et al., 1978; Calinescu et al., 2011; Goldengorin, 2009; Buchbinder et al., 2017; Buchbinder & Feldman, 2018), under cardinality or matroid constraints. Little is known on continuous set function optimization, and thus the tools, concepts, and strategies devised herein can be of further interest elsewhere.

3. Our approach is fully general in the sense that it applies to most random process data (including functional data, time series, and point processes). The functional parameters of the processes of interest (say mean functions, volatility functions, or intensity functions) are modelled by composing an inverse link function with a basis function representation, and Bayesian modeling is conducted via latent Gaussian models and inference is conducted via INLA (Integrated Nested Laplace Approximations) (Rue et al., 2009; 2017).

4. An extension of the proposed method applies also to the context where the interest is on comparing more than one marginal feature via a multi-objective version of the set function optimization problem of interest.

**Related prior work**. While the learning problem introduced above is new—and while there are novel contributions that will arise from our solution to it—there are some recent approaches in the context of functional data analysis (Ramsay & Silverman, 2002; 2006; Ferraty & Vieu, 2006; Horváth & Kokoszka, 2012) that are tangentially related to it, and that are briefly reviewed below.

Pini & Vantini (2016; 2017) propose an interval testing procedure for functional data that points out specific differences between functional populations. Also in the context of functional data, Berrendero et al. (2016) propose a discretization method consisting on learning to choose a finite collection of points in the domain of a set of functions in order to improve the performance of a functional data classifier. Martos & de Carvalho (2018) propose a Mann–Whitney type of statistic for functional data to learn about the regions at which two processes differ the most on aspects related with symmetry. Finally, Dette & Kokot (2021) develop hypothesis tests for the equivalence of functional parameters in a two sample functional data setup.

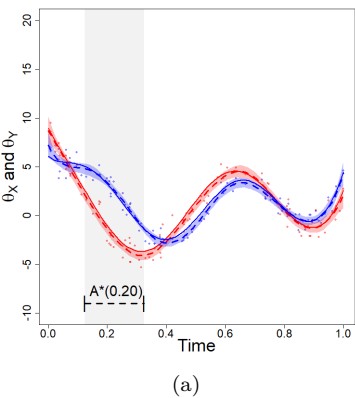 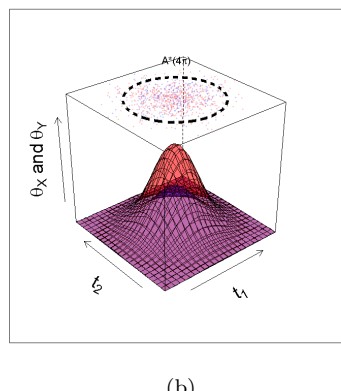 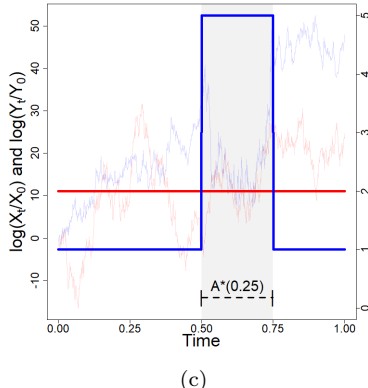

(a)          (b)          (c)

Figure 1: Sets of maximum dissimilarity for Examples 1–3. The sets are depicted using black dotted segments in the case of mean functions and volatility functions (a, c), and on the top of the box with a black dotted line in the case of intensity functions (b). Raw data alongside the true and estimated functional parameters $\theta_X$ and $\theta_Y$ are depicted in red and blue respectively; the proposed inferences for the sets will be discussed in Section 2.2.

Our approach differs from the ones mentioned above in a number of important ways. Perhaps the most important ones are that: $i$) here the goal is not to test hypothesis but rather to learn about regions with a given volume where two processes differ the most; $ii$) our approach applies to random processes in general whereas the methodologies reviewed above have mainly been designed with the context of functional data analysis in mind.

**Outline**. In Section 2 we introduce sets of maximum dissimilarity, present examples, and introduce the inference methods, whereas in in Section 3 we comment on extensions. In Section 4 we showcase the proposed methods on real data applications. Closing remarks are given in Section 5. Proofs and numerical experiments with artificial data are presented in the Appendix. Table 1 in the Appendix lists the main symbols and notation used throughout the article. We write $X(t)$ instead of $X_t$ when typographically convenient.

## 2 Learning About Sets of Maximum Dissimilarity

### 2.1 Sets of Maximum Dissimilarity

**Instances**

Prior to introducing sets of maximum dissimilarity in a formal fashion, we introduce some simple instances of the concept based on different parameter functions $\theta_X$ and $\theta_Y$. The respective sets of maximum dissimilarity and parameter functions for Examples 1–3 to be presented below are depicted in Fig. 1. The 'size' of these sets ($c > 0$) should be specified by the user based on the desired level of granularity for the analysis; this enables tracking of maximum differences on a weekly, monthly, or annual basis, for example. This will become clearer through the examples below and the data applications in Section 4.

**Example 1** (Mean Functions). Let $X_t = \theta_X(t) + \varepsilon_X(t)$ and $Y_t = \theta_Y(t) + \varepsilon_Y(t)$ be two stochastic processes defined on the unit interval, $T = [0, 1]$, with

$$\begin{cases} \theta_X(t) = E(X_t) = b(t) + 4\cos(10t) - 2(t - 0.75)^2, \\ \theta_Y(t) = E(Y_t) = b(t) + 3\sin(12t), \end{cases} \tag{2}$$

where $b(t) = 1/2 \exp\{10(t - 0.5)^2\}$ is a baseline curve, and $\varepsilon_X(t)$ and $\varepsilon_Y(t)$ are zero mean Gaussian error functions. Fig. 1(a) shows the connected set of size $c = 0.2$ where both mean functions differ the most in the $L^2$ sense.

**Example 2** (Intensity Functions)**.** Consider the intensity functions associated to two point processes

$$\begin{cases} \theta_X(t) = \lambda_X(t) = \gamma \exp\{-(t_1^2 + t_2^2)/2\}, \\ \theta_Y(t) = \lambda_Y(t) = \delta\theta_X(t), \end{cases} \tag{3}$$

defined over the region $T = [-3, 3]^2$ with $\gamma, \delta > 0$. Fig. 1(b) shows the true ball of maximum dissimilarity with area $4\pi$ at which the two intensity functions differ the most in the $L^2$ sense in the case where $\gamma = 100$ and $\delta = 1/2$.

**Example 3** (Volatility Functions)**.** Suppose $X_t$ and $Y_t$ are the log returns of two stock markets, with $E(X_t) = E(Y_t) = 0$, and that the goal is to search for the period of about a quarter (think of $T = [0, 1]$, so that $c = 0.25$), where the volatility between both markets differed the most. Let

$$\begin{cases} \theta_X(t) = \sigma_X(t) := \{\text{var}(X_t)\}^{1/2}, \\ \theta_Y(t) = \sigma_Y(t) := \{\text{var}(Y_t)\}^{1/2}. \end{cases}$$

Fig. 1(c) depicts a simulated example of two artificial stock prices evolving during a period of 1 year ($T = [0, 1]$), along with the 3 month period over which the volatilities in both markets differed the most in the $L^2$ sense.

To allow for visualizations, in the examples above we focused on low-dimensional instances but the theory to be presented next holds in general for any compact ground set $T \subset \mathbf{R}^d$.

**Regions of maximum dissimilarity**

Let $\theta_X \equiv \theta_X(t)$ and $\theta_Y \equiv \theta_Y(t)$ be functional parameters characterizing $q$ different marginal features of the processes $X$ and $Y$ for $t \in T \subset \mathbf{R}^d$. Throughout, we will assume that the ground set $T$ is compact and that $\theta_X$ and $\theta_Y$ live in the Banach space $(L^p(T), \|\cdot\|_p)$, with $\|f\|_p = (\int_T |f|^p \, d\mu)^{1/p}$, where $\mu$ is a measure over the Borel sets on $T$; we will refer to $\|f\|_p^{(A)} = (\int_A |f|^p \, d\mu)^{1/p}$ as the $L^p$ sub–norm over $A \subseteq T$. We equip the set $\mathcal{A}$ of compact and convex subsets of the ground set $T$, with the Hausdorff distance

$$d_{\text{H}}(A, B) = \max\left\{\max_{t \in A} d(t, B), \max_{t \in B} d(t, A)\right\}, \qquad A, B \in \mathcal{A}.$$

where $d(t, B)$ is the minimum Euclidean distance from the point $t \in A$ to the set $B$.

A compact and convex set $A \subseteq T$ is said to be a *region of maximum dissimilarity* (in the $L^p$ sense), *with volume bounded by* $c > 0$, if it maximizes $F(A) = \|\theta_X - \theta_Y\|_p^{(A)}$ and if its volume, $|A|$, does not exceed $c$; the formal definition is as follows.

**Definition 1** ($L^p$ Region of Maximum Dissimilarity)**.** *Suppose $\theta_X, \theta_Y \in L^p(T)$ and $T \subset \mathbf{R}^d$ is compact. A region of maximum dissimilarity (RMD) is defined as a set $A_c^* \subseteq T$ that solves,*

$$\begin{aligned} \max_{A \subseteq T} \quad & \|\theta_X - \theta_Y\|_p^{(A)} \\ s.t. \quad & |A| \leq c, \\ & A \text{ is compact and convex,} \end{aligned} \tag{4}$$

*for a fixed $c \geq 0$. In addition, $D_c^*(\theta_X, \theta_Y) = \|\theta_X - \theta_Y\|_p^{(A_c^*)}$ is said to be the dissimilarity index.*

The optimization problem in (4) is a continuous set function optimization problem similar to (1), with

$$\begin{cases} F(A) = \|\theta_X - \theta_Y\|_p^{(A)}, \\ \mathscr{A} = T, \\ \mathscr{F}_c = \{A : |A| \leq c, A \text{ is compact and convex}\}. \end{cases}$$

Clearly, $F(A)$ is an increasing set function, that is, if $A \subseteq B$ then $F(A) \leq F(B)$. Also, $F(A)$ is modular (or additive) in the sense that $F(A) + F(B) = F(A \cup B) + F(A \cap B)$. While it is evident that $A_c^*$ exists for a few specific examples (say, $A_{|T|}^* = T$ almost everywhere, if $T$ is convex), the existence of $A_c^*$ in general is not straightforward and it is proved below in Theorem 1. RMDs and dissimilarity indices also have a number of attributes which we summarize over Theorem 1.

**Theorem 1.** *Suppose $\theta_X, \theta_Y \in L^p(T)$ and $T \subset \mathbf{R}^d$ is compact. The quantities $A_c^*$ and $D_c^*$ obey the following properties:*

1. *$A_c^*$ exists, for every $c \geq 0$.*

2. *Suppose $\mu(A) = |A|$. Then, the RMDs of $(K\theta_X, K\theta_Y)$ and $(L\theta_X, L\theta_Y)$ are respectively $A_c^K = A_c^*$ and $A_c^L = A_{\alpha+\beta c}^*$, where $K\theta(t) = \alpha + \beta\theta(t)$, $L\theta(t) = \theta(\alpha + \beta t)$, $\alpha \in \mathbf{R}$, and $\beta \neq 0$.*

3. *$D_c^*(\theta_X, \theta_Y)$ is a distance over $\Theta_c = \{\theta|_A : \theta \in L^p(T), A \in \mathscr{F}_c\}$, for $c > 0$.*

4. *$D_c^*(\theta_X, \theta_Y)$ is non-decreasing as a function of $c$, for fixed $\theta_X, \theta_Y \in L^p(T)$.*

Theorem 1 warrants some comments. The existence of RMDs (Claim 1) follows from the continuity of the volume functional, $|A|$, and the compactness of the search domain, $\mathscr{F}_c$; see Appendix A. An RMD needs not however to be unique, as can be easily seen by considering the limiting case $\theta_X = \theta_Y$ for which every compact and convex subset of $T$ with measure $c$ is an RMD. Claim 2 shows how RMDs are impacted by a group of linear transformations acting either on the functional parameter or over time; the assumptions $\mu(A) = |A|$ and $\beta \neq 0$ are only required for $A_c^L = A_{\alpha+\beta c}^*$, as indeed $A_c^K = A_c^*$ holds more generally. Claim 3 implies that dissimilarity indices have a metric interpretation. Finally, Claim 4 notes that $D_c^* = D_c^*(\theta_X, \theta_Y)$ cannot decrease as $c$ increases.

For practical considerations entailed in the estimation of $A_c^*$ from data (more details in Section 2.2), next we introduce the $L^p$ balls of maximum dissimilarity as a suitable parametric approximation to RMDs.

**Balls of maximum dissimilarity**

Let $\mathscr{B}_p$ be the family of all closed balls in $L^p(T)$, for $p \geq 1$, that is,

$$\mathscr{B}_p = \{B(t,r) : r > 0, t \in T\},$$

where $B(t,r) = \{s : \|s - t\|_p \leq r\}$; to ease notation we drop the dependence of $B(t,r)$ on $p$. A more parsimonious option for modeling and computing is to put more structure on the shape of RMDs, and this leads to the next definition.

**Definition 2** (*$L^p$ Ball of Maximum Dissimilarity*). *Suppose $\theta_X, \theta_Y \in L^p(T)$ and $T \subset \mathbf{R}^d$ is compact. An $L^p$ ball of maximum dissimilarity (BMD) is defined as a set $B_c^* \subseteq T$ that solves,*

$$
\begin{aligned}
\max_{B \subseteq T} \quad & \|\theta_X - \theta_Y\|_p^{(B)} \\
s.t. \quad & |B| \leq c \\
& B \in \mathscr{B}_p,
\end{aligned}
\tag{5}
$$

*for a fixed $c \geq 0$.*

As in Definition 5 we refer to $\mathscr{D}_c^* = \mathscr{D}_c^*(\theta_X, \theta_Y) = F(B_c^*)$ as the dissimilarity index for BMDs. Since the volume of an $L^p(T)$ ball of radius $r$ in $\mathbf{R}^d$ is

$$|B(t,r)| = \frac{\{2r\Gamma(1/p + 1)\}^d}{\Gamma(d/p + 1)},$$

for all $t \in T$, where $\Gamma(z) = \int_0^\infty x^{z-1}e^{-x}\,dx$ is the Gamma function, it follows that the volume constraint on BMDs, $|B| \leq c$, can be rewritten as a function of the radius, that is

$$r \leq \frac{\{c\Gamma(d/p + 1)\}^{1/d}}{2\Gamma(1/p + 1)} \equiv R_c. \tag{6}$$

Equation (6) is computationally convenient and will be revisited in further detail in Section 2.2.

Given the similarities between the definitions of RMDs and BMDs it is not surprising that they have identical properties (e.g., for $c > 0$, $\mathscr{D}_c^*(\theta_X, \theta_Y)$ is a also distance and $\mathscr{D}_c^*$ is non-decreasing). More generally, the following theorem holds:

**Theorem 2.** *Suppose $\theta_X, \theta_Y \in L^p(T)$ and $T \subset \mathbf{R}^d$ is compact. The quantities $B_c^*$ and $\mathscr{D}_c^*$ obey the following properties:*

1. *$B_c^*$ exists, for every $c \geq 0$.*

2. *$\mathscr{D}_c^*$ is continuous, and the "argmax" correspondence of center–radius, $\alpha_c : [0, \infty) \twoheadrightarrow T \times [0, R_c]$, defined as $\alpha_c = \{(t, r) \in T \times [0, R_c] : f(t, r) = \mathscr{D}_c^*\}$, is upper hemicontinuous for every $c \geq 0$, where $f(t, r) = F\{B(t, r)\}$.*

Theorem 2 proves the existence of BMDs and notes that the smoothness of $R_c$ is inherited by $\mathscr{D}_c^*$.

Next, we switch gears and focus on learning about BMDs from data.

## 2.2 Learning About BMDs from Data

### Computing and optimization

As anticipated earlier, BMDs are computationally convenient as a suitable parametric approximation to RMDs. This becomes apparent from (5) as finding an optimal ball corresponds to a standard constrained optimization problem, where the objective function depends solely on the center–radius vector with $d + 1$ components. In addition, (6) highlights another computational advantage of modeling RMDs via BMDs: The volume constraint can be directly expressed as a function of one of the arguments of the objective function, specifically the radius. To put it differently, the key takeaway from (5) and (6) is that the set function constrained optimization problem leading to BMDs can actually be formulated as a standard continuous optimization problem over $T \times [0, R_c] \subset \mathbf{R}^{d+1}$. Indeed, it follows from (5) and (6) that computing a BMD is equivalent to solving

$$\max\{f(t, r) : (t, r) \in T \times [0, R_c]\}, \tag{7}$$

where $f(t, r) = F\{B(t, r)\}$. When the center of the optimal BMD ($t_c^*$) is 'sufficiently far' from the boundary of $T$, the optimal radius ($r_c^*$) is $R_c$ in (6). That is, when $d(t_c^*, \partial T) > R_c$ the optimal radius is $r_c^* = R_c$ (as $f(t, r)$ is a non-decreasing function of $r$) in which case the optimization problem in (7) resumes to searching for $t_c^* \in \mathbf{R}^d$. This also implies that often in practice the marginal posterior of $r_c^*$ is essentially degenerated.

### Latent Gaussian model specification

To model BMDs in applications we consider a version of the latent Gaussian model specification in Rue et al. (2009) adapted to our setup; to ease notation, below we only refer to $X(t)$, and denote its functional parameter by $\theta(t) \equiv \theta_X(t)$, but all comments apply to $Y(t)$ as well. A latent Gaussian model is essentially a Bayesian generalized additive model that assigns Gaussian priors to parameters and a possibly non-Gaussian prior to its hyperparameters. Specifically, suppose that $Z(t) = h\{X(t)\}$ is in the exponential family, with its mean function coinciding with the functional parameter, and that

$$\theta(t) = g\left(\beta_0 + \sum_{i=1}^{B} \beta_i \phi_i(t)\right). \tag{8}$$

Here $\{\phi_i \equiv \phi_i(t)\}_{i=1}^{B}$ is a set of basis functions in $L^p(T)$, $\beta = (\beta_0, \ldots, \beta_B)^{\mathrm{T}}$ is a parameter, and $g$ is an inverse link function. Following Rue et al. we assign a multivariate Normal prior with a sparse precision matrix ($Q$) to $\beta$, which induces a Gaussian process prior on $g^{-1}(\theta(t))$ with a conditional independence property. Many functional parameters can be modeled in this way including those from Examples 1–3 and all numerical instances in Section 4 and Appendix C. The theoretical developments from Section 2.1 apply however more generally beyond the modeling assumptions made over this section.

### INLA-based inference for balls of maximum dissimilarity

We now discuss how to conduct inference for balls of maximum dissimilarity. It is well-known that the latent Gaussian model described above can be fitted with an Integrated Nested Laplace Approximation (INLA)

(Rue et al., 2009); the method is effective even when the dimension of the precision matrix $Q$ is large, and is particularly tailored for the case where the number of hyperparameters, $\alpha$, is moderate (say, 6–12). INLA is a deterministic method for approximating the marginal posterior of each parameter that is based on the Laplace approximation; loosely speaking, the Laplace approximation is an approximation for integrals of the type $\int e^{-nf(y)}\,\mathrm{d}y$ for large $n$, that approximates the integrand $(e^{-nf(y)})$ with a Gaussian density centered at its mode and sets the covariance matrix as the inverse of the curvature (around the mode); see, for instance, Young & Smith (2005, Section 9.7). Below, we sketch some brief details on INLA. The first step of INLA approximates the marginal posterior of $\alpha$ via the Laplace approximation, that is,

$$p(\alpha \mid \mathrm{data}) = \frac{p(\alpha, \beta \mid \mathrm{data})}{p(\beta \mid \alpha, \mathrm{data})} \approx \left. \frac{p(\alpha, \beta \mid \mathrm{data})}{\widetilde{p}(\beta \mid \alpha, \mathrm{data})} \right|_{\beta = \beta_\alpha^*}, \tag{9}$$

where $\widetilde{p}(\beta \mid \alpha, \mathrm{data})$ is the Gaussian approximation based on the mode of the full conditional of $\beta$, and where $\beta_\alpha$ is the mode of this approximated full conditional for a given $\alpha$. Next, INLA approximates the marginal posterior of each component of $\beta$. Let $\beta_{-i}$ be the elements of $\beta$, except $\beta_i$. Similarly to (9) it follows that

$$p(\beta_i \mid \alpha, \mathrm{data}) \propto \frac{p(\alpha, \beta \mid \mathrm{data})}{p(\beta_{-i} \mid \alpha, \beta_i, \mathrm{data})}, \tag{10}$$

which can be approximated using a Laplace approximation for $p(\beta_{-i} \mid \alpha, \beta_i, \mathrm{data})$. Finally, the marginal posterior density $p(\beta_i \mid \mathrm{data})$ is obtained by numerically integrating out $\alpha$. Below, INLA is applied to the data from processes $X$ and $Y$ as $\mathcal{D}_X$ and $\mathcal{D}_Y$. Estimation and inference for BMDs can be conducted using Algorithm 1 that combines the deterministic nature of INLA along with sampling.

---

**Algorithm 1** INLA-based posterior inference for BMDs

---

**Step 1:** Fit the marginal posterior densities, $p(\beta_{X,i} \mid \mathcal{D}_X)$ and $p(\beta_{Y,i} \mid \mathcal{D}_Y)$, using the Integrated Nested Laplace Approximation, for $i = 0, \ldots, B_X$ and $j = 0, \ldots, B_Y$.

**Step 2:** Sample $m$ posterior draws from the full posterior of $\beta_X = (\beta_{X,0}, \ldots, \beta_{X,B_X})$ and $\beta_Y = (\beta_{Y,0}, \ldots, \beta_{Y,B_Y})$, given by

$$\beta_X^{(1)}, \ldots, \beta_X^{(m)} \overset{\mathrm{iid}}{\sim} p(\beta_X \mid \mathcal{D}_X), \qquad \beta_Y^{(1)}, \ldots, \beta_Y^{(m)} \overset{\mathrm{iid}}{\sim} p(\beta_Y \mid \mathcal{D}_Y),$$

to generate $m$ posterior trajectories from the functional parameters using (8); that is, for $k = 1, \ldots, m$, do

$$\theta_X^{(k)}(t) = g\left( \beta_{X,0}^{(k)} + \sum_{i=1}^{B_X} \beta_{X,i}^{(k)} \phi_i(t) \right), \qquad \theta_Y^{(k)}(t) = g\left( \beta_{Y,0}^{(k)} + \sum_{j=1}^{B_Y} \beta_{Y,j}^{(k)} \phi_j(t) \right). \tag{11}$$

**Step 3:** Use the posterior trajectories from Step 2 to obtain a sequence of posterior BMD draws for

$$\{ B_c^{(k)*} \equiv B(t_c^{(k)*}, r_c^{(k)*}) \}_{k=1}^m,$$

with $(t_c^{(k)*}, r_c^{(k)*})$ solving the optimization problem in (7), for $k = 1, \ldots, m$.

---

Algorithm 1 warrants some comments. Step 1 is deterministic, it follows by numerically integrating out the hyperparameters in (10), and to facilitate its implementation we recommend using the `R-INLA` package (Martins et al., 2013; Lindgren et al., 2015) from `R` (R Development Core Team, 2022), which is also equipped with routines that facilitate the implementation of Step 2 (e.g., `inla.posterior.sample`). Step 3 boils down to solving the optimization problem in (7) using the pair of posterior trajectories in (11) ; thus we solve (7) per each INLA iteration and the posterior quantifies the uncertainty surrounding the optimal centre and radius. Except where mentioned otherwise, in all experiments reported below, we draw $m = 1\,000$ times from the posterior distribution of BMDs using Algorithm 1.

## 3 Multi-Objective RMDs

We now consider the context where the interest is on learning about regions over which a set of marginal features of two processes most differ. Let

$$\begin{cases} \theta_X = (\theta_{X,1}, \dots, \theta_{X,q}) = (\theta_{X,1}(t), \dots, \theta_{X,q}(t)), \\ \theta_Y = (\theta_{Y,1}, \dots, \theta_{Y,q}) = (\theta_{Y,1}(t), \dots, \theta_{Y,q}(t)), \end{cases}$$

be functional parameters characterizing marginal features of these processes. In this section

$$F_i(A) = \|\theta_{X,i} - \theta_{Y,i}\|^{(A)}, \tag{12}$$

denotes the set objective functions of interest, for $i = 1, \dots, q$. Ideally, we would aim to simultaneously maximize all targets $(F_1(A), \dots, F_q(A))$. Yet, in practice targets may conflict each other, that is, if one is increased some other may be decreased. The following Pareto optimal concept induces an order over the collection of feasible sets $\mathscr{F}_c$, and it defines optimality via a compromise across all objective set functions, in the sense that improvements on one target cannot be made at the cost of deteriorating another target.

**Definition 3** (Pareto Optimal Region of Multi-maximum Dissimilarity). *Let $F_i(A)$ be defined as in* (12). *The set $A_c^* \in \mathscr{F}_c$ is a Pareto optimal region of multi-maximum dissimilarity if there exists no other set $A \in \mathscr{F}_c$ such that $F_i(A) \geq F_i(A_c^*)$, for all $i \in \{1, \dots, q\}$, and $F_i(A) > F_j(A_c^*)$, for at least one $j \in \{1, \dots, q\}$.*

In common with standard theory for multi-objective function, in our set function context the set of Pareto optimal RMMDs can be analytically obtained only in very specific cases, and thus we need to resort to scalarization (Pardalos et al., 2017, Ch. 2) to reduce a multi-objective problem into a single-objective problem. Here, we define and characterize the following linear scalarization method for our set function context

$$\mathcal{F}_w(A) = \sum_{i=1}^{q} w_i F_i(A), \tag{13}$$

$w = (w_1, \dots, w_q) \in (0, \infty)^q$, and we will refer to

$$A_{w,c}^* = \arg \max_{A \in \mathscr{F}_c} \mathcal{F}_w(A), \tag{14}$$

as the solution to the set function linear scalarization problem with weight $w$.

**Theorem 3.** *Every solution to the linear scalarization problem is a Pareto optimal RMMD.*

Section 4.2 illustrates how Theorem 3 can be applied in practice.

## 4 Empirical Section

In Appendix C we assess the performance of the proposed tools via a Monte Carlo study. Next, we showcase real data applications.

### 4.1 Thefts in Buenos Aires

Buenos Aires is the most dense metropolis in Argentina and its crime rates are significantly higher in comparison to the rest of the country. In this section we illustrate how the proposed method can reveal regions of the city where nonviolent crimes—such as burglary, pickpocketing or nonviolent thefts—have changed the most, comparing the years 2019 (pre COVID-19) and 2020 (when the COVID lockdown took place during several months). The data are publicly available online in the city hall web page (*https://data.buenosaires.gob.ar*), and consist of point process data on the latitude and longitude where thefts occurred during 2019 ($\mathcal{D}_{2019}$) and 2020 ($\mathcal{D}_{2020}$). Here, the functional parameters of interest are the intensity functions

$$\theta_{2019}(\text{latitude}, \text{longitude}), \qquad \theta_{2020}(\text{latitude}, \text{longitude}),$$

and its BMD will represent region of the city, of a given size, where the most noteworthy changes in thefts took place. The fitted BMDs were modeled according to (8) using a log-Gaussian Cox process with a similar prior specification as in Appendix C. In Fig. 2(a) we depict the estimated BMD corresponding to an area of 8km² along with an heat map of the differences in the estimated posterior intensity functions between consecutive years; the value of 8km² was chosen for illustration as it corresponds to about twice the size of the largest neighborhood, which is Palermo. We also depict in Fig. 2(b) an heat map of the posterior density corresponding to the center of the BMD which shows that these are substantially concentrated, thus suggesting low uncertainty on the fitted BMD.

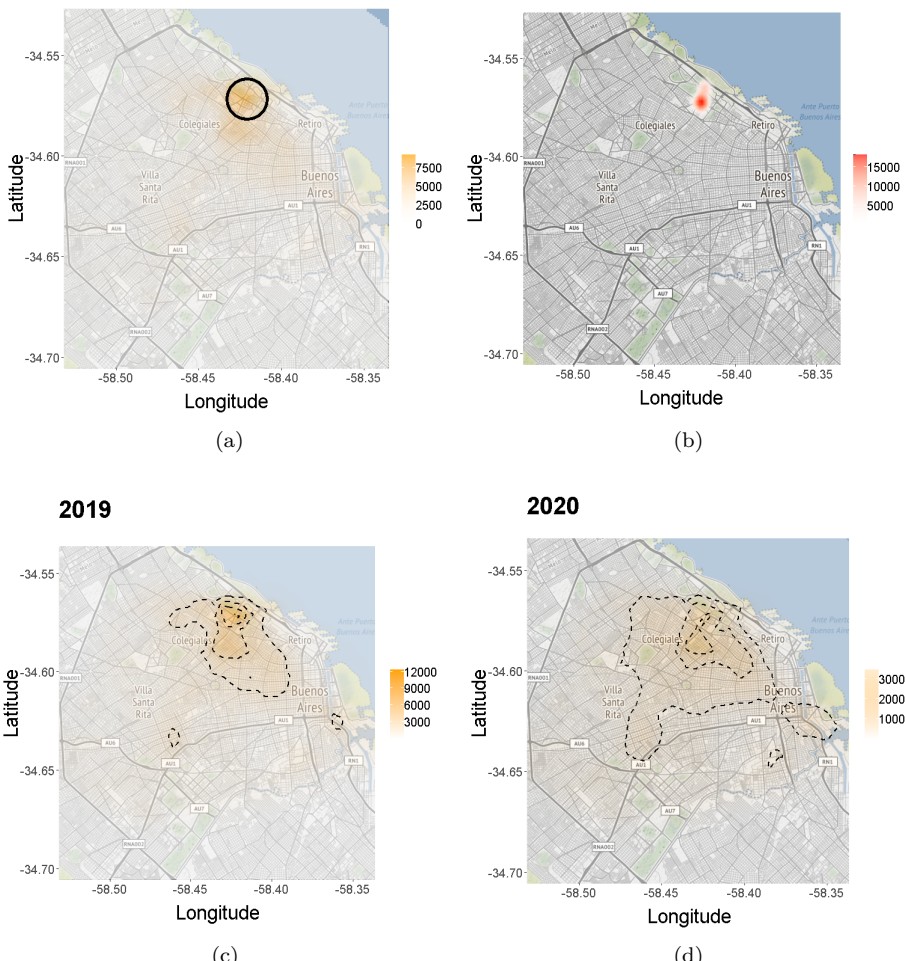

Figure 2: (a) Fitted BMD corresponding to $c = 8$km² along with a contour of pointwise differences between fitted posterior intensity functions over consecutive years. (b) Posterior density estimation corresponding to the centers of of the fitted BMD. (c) and (d) shows the fitted intensity functions for 2019 and 2020.

To clarify the applied meaning of such BMDs we provide some additional background on the social context surrounding the empirical analysis. In Fig. 2(c–d) we depict the fitted intensity functions corresponding to both years. As can be seen from Fig. 2, in 2019 and 2020 thefts were more likely to occur in APRV (Almagro, Palermo, Recoleta, and Villa Crespo) which are some of the neighborhoods where several commercial and touristic activities took place. Yet, important differences on the estimated intensity functions are perceived between both years. During the first half of year 2020, local authorities took strong social distancing measures such as the limitation to the access the public transportation system, restrictions on business and commerce

during the day, limitations on gatherings and tourism activities, restriction to the capacity in bars and restaurants, among others. The difference on the estimated intensity functions between consecutive years evident from Fig. 2—and the implied reduction of thefts over 2020—is in line with the findings of Mohler et al. (2020) that report similar evidence on the effect of COVID-19 lockdown and social distance policies in nonviolent crime. The BMD in Fig. 2(a) suggests that APRV (Almagro, Palermo, Recoleta, and Villa Crespo) are the neighborhoods where there was a most impactful effect of COVID-19 lockdown. To put it differently, while nonviolent crime has decreased during lockdown over the entire city, what the fitted BMD highlights is that such reduction was relatively much higher in the APRV neighborhoods.

## 4.2 Volatility in Stock Markets

Our second illustration will shed light on the multi-objective approach from Section 3. Data were gathered from Yahoo Finance and consist of monthly values of the NYSE and NASDAQ composite indices from the New York Stock Exchange ($\{X_t\}$) and the NASDAQ Stock Exchange ($\{Y_t\}$), respectively. The data ranges from January 1990 to May 2021, thus covering a variety of episodes of financial turbulence such as the dotcom tech bubble that peaked around 2000, the subprime crisis that started around 2007, and the recent COVID–19 global pandemic. In the multi-objective RMD analysis to be conducted here, we will consider two functional parameters of interest: The mean values of the indices over time and the volatility of their log returns, that is,

$$\begin{cases} m_X(t) = E(X_t), \\ m_Y(t) = E(Y_t), \end{cases} \qquad \begin{cases} \sigma_X(t) = [E\{\log(X_t/X_{t-1})^2\}]^{1/2}, \\ \sigma_Y(t) = [E\{\log(Y_t/Y_{t-1})^2\}]^{1/2}. \end{cases}$$

These functional parameters were modeled according to (8), using an identity link function and B-spline basis functions, choosing the number of basis functions using the DIC (Deviance Information Criterion; Spiegelhalter et al. 2002; 2014), and using an uninformative prior.

We consider intervals of six months, one year, and two years (corresponding to $c = 6, 12$, and 24) and we aim to evaluate on what periods of such length these two stock markets differed the most—in terms of both average returns as well as volatility. We thus seek for the interval of time $B_c^* = [t_c^* - r_c^*, t_c^* + r_c^*]$ that as in (14) maximizes the following scalarized set function optimization problem,

$$\max\{\mathcal{F}_w\{B(t,r)\} : (t,r) \in T \times [0, R_c]\}, \tag{15}$$

where $w$ is the scalarization parameter and

$$\mathcal{F}_w\{B(t,r)\} = w \int_{t-r}^{t+r} |m_X(u) - m_Y(u)| \, \mathrm{d}u + (1-w) \int_{t-r}^{t+r} |\sigma_X(u) - \sigma_Y(u)| \, \mathrm{d}u.$$

It follows from Theorem 3 that every solution to the linear scalarization problem (15) is a Pareto optimal BMMD (ball of multi-maximum dissimilarity), and hence Pareto optimal BMMDs obtained by linear scalarization have the nice feature of allowing one to put more emphasis on the mean values or on volatilities according to how we set $w$. That is, by setting $w = 0$ or $w = 1$, we only consider volatilities or mean values respectively and the analysis corresponds to a standard BMD, while for $w \in (0, 1)$ absolute values in differences between mean functions are more important than those in volatilities as $w$ increases. In terms of computing we adapt Algorithm 1 to the multi-objective context. That is, inference about the BMMDs is conducted by sampling $m = 1\,000$ times from the posteriors for means and volatilities—and rather than solving (7) as in Algorithm 1—we now solve the scalarized set function optimization problem in (15).

In Fig. 3 we compare the results of the single BMD analysis [panels (a) to (d)] versus the multi-objective analysis [panels (e) and (f)] considering a value of $c$ corresponding to a period of six months, one year, and two years. As can be seen from Fig. 3(a–b), the BMD associated with the mean ($w = 1$) concentrates around the subprime crisis, indicating that July 2006 to July 2008 is the period over which mean levels of NYSE and NASDAQ differed the most. In Fig. 3(c–d), we see that the BMD associated with volatility ($w = 0$) ranges from October 1999 to October 2001—which corresponds to the dotcom bubble burst. Finally, the multi-objective approach with $w = 2/3$ is depicted in Fig. 3 (e–f) and it suggests that volatility has a greater

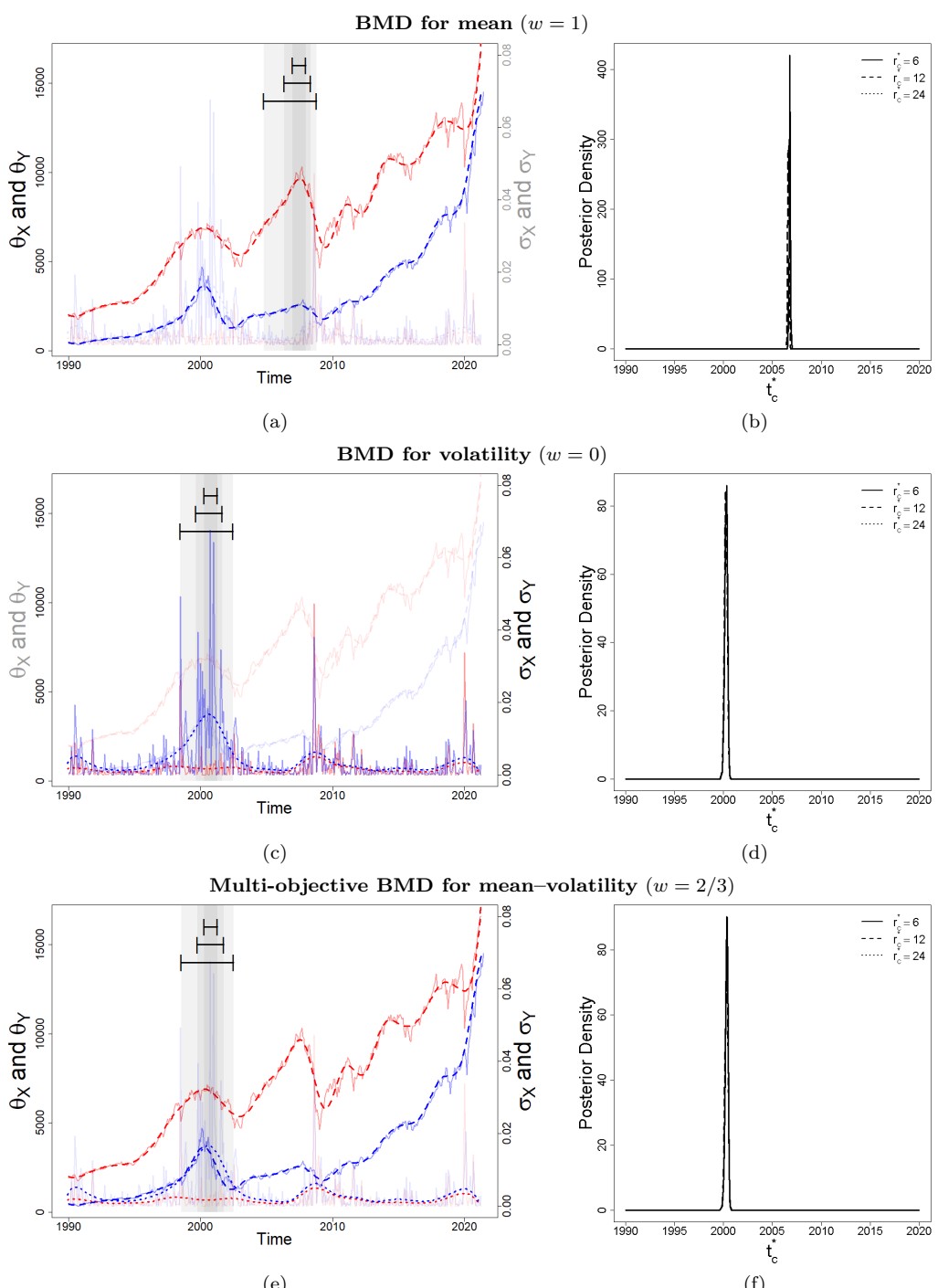

Figure 3: (a, c, e) Black solid segments corresponds to fitted multi-objective BMDs for six monts, one year, and two years for $w = 1$, $w = 0$, and $w = 2/3$; the charts also show the realized (solid) and estimated (dotted) volatilities corresponding to NYSE (red) and NASDAQ (blue). (b, d, f) Marginal posterior density for the centre of the BMD.

control on the objective function in (15); that is, even when we set $w = 2/3$—that is, even when we set more emphasis on the differences in means rather than the differences in volatility—we still get a similar

result as setting $w = 0$, as we end up recovering the period of the dotcom bubble burst as can be seen from Fig. 3(e–f).

### 4.3 Electrocardiogram Data (ECG200)

For our final illustration we use the ECG200 dataset contributed by Olszewski (2001). The data is the result of monitoring electrical activity recorded during one heartbeat and it consists of 200 ECG signals sampled at 96 time instants, corresponding to 133 normal heartbeats ($\mathcal{D}_X$) and 67 myocardial infarction signals ($\mathcal{D}_Y$); the data are publicly available from the UCR Time Series Classification and Clustering website (*http://www.cs.ucr.edu/~eamonn/time_series_data_2018/*). In this illustration, the functional parameters of interest are the mean functions of ECG signals for both classes (normal heartbeat $\theta_X(t) = \mathrm{E}(X_t)$, and myocardial infarction $\theta_Y(t) = \mathrm{E}(Y_t)$) and one of the goals of the analysis is to track down periods, of a given length, over a cardiac cycle where the differences between the two classes is most pronounced. To model these functional parameters the Gaussian process prior specification in (8) was once more applied using an identity link, B-spline basis functions, the DIC to select the number of basis functions, and a Matérn covariance function with the PC prior of Fuglstad et al. (2019) setting $P(\sigma > 1) = 0.001$ and $P(\ell < 0.05) = 0.001$.

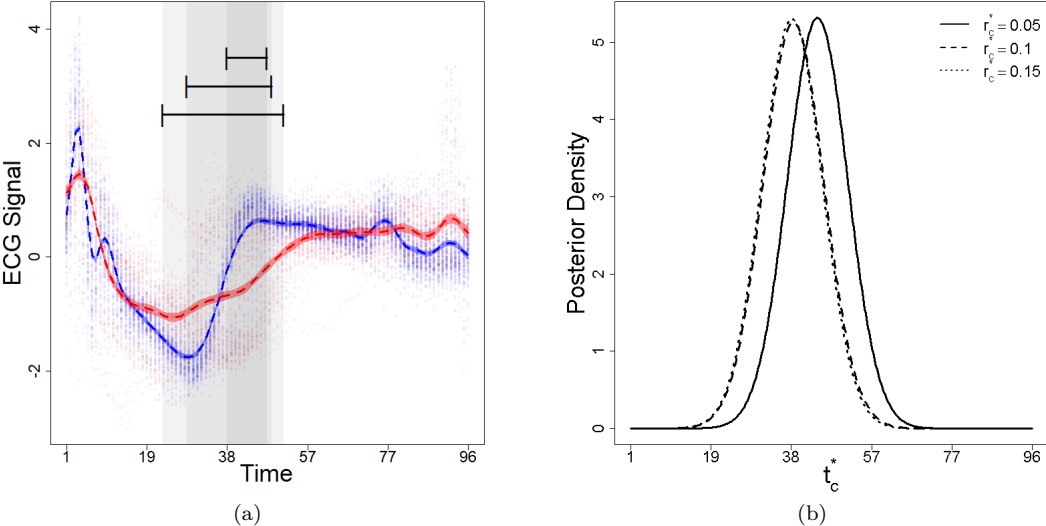

(a)                                                          (b)

Figure 4: ECG200 dataset: (a) Black solid segments correspond to fitted BMDs, $\widehat{B}_{10}^* \subset \widehat{B}_{20}^* \subset \widehat{B}_{30}^*$; the chart also shows the raw data (dots) and fitted mean functions (dashed) along with 95% credible bands, where red and blue respectively correspond to normal heartbeats and myocardial infarction signals. (b) Marginal posterior density for the centre of the set of maximum dissimilarity $B_{10}^*$.

In Fig. 4(a), we depict the fitted posterior estimates for a sequence of BMDs ($\widehat{B}_{10}^*$, $\widehat{B}_{20}^*$, and $\widehat{B}_{30}^*$) using black solid segments, along with the raw data, and the fitted mean functions with 95% credible bands. The obtained BMDs uncover periods of a given length where we observe largest differences between the estimated mean ECG functions. Informally, we may think of such BMDs as corresponding to intervals of about 10–30 deciseconds, since the 96 time instants cover about a cardiac cycle for all subjects and thus are not expected to last longer than 1 second. The fitted BMD centers are localized around the time instants 22–55. All in all, the analysis suggests that while normal heartbeats and myocardial infarction signals have similar 'peaks' at the beginning of the sample period (i.e. they have similar Q waves, in ECG signal analysis terminology), immediately in the period right after (i.e. over their ST segments) they greatly differ.

We close the analysis with two final remarks. First, in this illustration the fitted BMDs verify the chained inclusions $\widehat{B}_{10}^* \subset \widehat{B}_{20}^* \subset \widehat{B}_{03}^*$, but this property does not hold in general—nor for the fitted BMDs, nor for the true ones; counterexamples can be constructed either numerically or analytically. Second, although BMDs are unrelated to classification, since the ECG200 dataset is a popular benchmark for new classifiers it

may be sensible to ask whether the accuracy of some classifiers at discriminating outcome classes (diseased–nondiseased) can be improved by focusing on BMDs rather than treating the entire time horizon equally; we leave such open problem for future analysis.

## 5 Discussion

Sets of maximum dissimilarity and their variants are here proposed as a tool for acquiring knowledge on the region with a given size where two stochastic processes differ the most. The proposed learning problem is shown to be equivalent to a continuous set function optimization on a monotone modular function, under a Lebesgue measure constraint. The existence of the proposed sets of maximum dissimilarity is nontrivial but we prove their existence, and illustrate with artificial and real data that it only requires a moderate computational investment to learn them from data. The proposed methods are developed in full generality for the setting where the data of interest are themselves stochastic processes, and thus the proposed toolbox can be used for unveiling the regions of maximum dissimilarity with a given volume, for a variety of random process data. All modeling was framed within a latent Gaussian framework, with inference being conducted using the Integrated Nested Laplace Approximation; clearly, other computational approaches could have been employed as well, including, for example, variational inference (Blei et al., 2017). A multi-objective version of the proposed framework is also devised to learn about multi-objective RMDs, where several functional parameters are considered—each characterizing a specific feature of the processes being compared.

While the theoretical developments from Section 2.1 establish the existence of general compact and convex sets of maximum dissimilarity, BMDs turn out be a much convenient simplification for a variety of reasons. First, numerical optimization is much more challenging with general RMDs, whereas for BMDs it is relatively simple as can be seen from (7). Second, the inference for general RMDs would entail averaging posterior simulated RMDs—that is, averaging sets—whereas for BMDs it suffices to compute the posterior mean of the $(d+1)$-dimensional centre-radius pair.

Some final comments on future research are in order. Firstly, while the paper focuses on scenarios where the user sets $c$, this approach could be integrated into a decision theory framework; although it remains an open problem for future analysis, Appendix E provides a hint on how this issue could be addressed. Secondly, this paper pioneers and introduces sets of maximum dissimilarity, which have been designed to highlight potential differences. Yet, in cases where $\{X_t\} = \{Y_t\}$, they may identify spurious regions. Assessing the statistical significance of these regions through multiple testing procedures seems conceivable, but the effectiveness of these approaches in this context requires further theoretical scrutiny and numerical analyses. Finally, discrete analogues of our problem may also be of interest in their own right, and may present themselves as a natural option for devising other practical algorithms for approximating continua RMD.

### Acknowledgments

We are grateful to the Action Editor and three anonymous reviewers for their suggestions and constructive feedback that have significantly improved the manuscript. We thank Vanda Inácio de Carvalho for comments and feedback and Finn Lindgren for discussions on INLA. MdC was partially supported by the Royal Society of Edinburgh and by FCT (Fundação para a Ciência e a Tecnologia, Portugal) under Grants https://doi.org/10.54499/UIDB/04106/2020 and https://doi.org/10.54499/UIDP/04106/2020.

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

## A    Auxiliary Lemmata

In this section we state some auxiliary results that will be used to prove the main results of this paper.

**Lemma 1** (Blaschke selection theorem). *The set of all compact convex subsets of a fixed compact subset of $\mathbf{R}^d$ is compact under the Hausdorff metric.*

**Lemma 2** (Berge's maximum theorem). *Let $\varphi : X \twoheadrightarrow Y$ be a continuous correspondence between topological spaces with nonempty compact values, and suppose that $g : Gr\, \varphi \to \mathbb{R}$ is continuous. Define the "value function" $v : X \to \mathbf{R}$ by*

$$v(x) = \max\{g(x,y) : y \in \varphi(x)\}$$

*and the correspondence $\alpha : X \twoheadrightarrow Y$ of maximizers by*

$$\alpha(x) = \{y \in \varphi(x) : g(x,y) = v(x)\}.$$

*Then:*

1. *The value function $v$ is continuous.*
2. *The "argmax" correspondence $\alpha$ has nonempty compact values.*
3. *If $Y$ is Hausdorff, then the "argmax" correspondence $\alpha$ is upper hemicontinuous.*

**Lemma 3** (Product of correspondences theorem). *The product of correspondences obeys the following properties:*

1. *The product of a family of upper hemicontinuous correspondences with compact values is upper hemicontinuous with compact values.*
2. *The product of a finite family of lower hemicontinuous correspondences is lower hemicontinuous.*

Lemma 1 is a celebrated result on convex analysis; the version above can be found, for instance, in Benyamini (1998). Lemmas 2 and 3 are well-known results on optimization of set-valued functions; the versions above can be found in Aliprantis & Border (2006, Theorems 17.31 and 17.28).

## B    Proofs of Main Results

### B.1    Proof of Theorem 1

**Claim 1**. We start by showing that
$$\mathscr{F}_c = \{A \in \mathcal{A} : |A| \le c\}$$
is compact, for every $c \ge 0$, where $\mathcal{A}$ is the family of compact and convex subsets of the ground set $T$. Recall that by the Blaschke selection theorem (Lemma 1), $\mathcal{A}$ is compact under the Hausdorff metric. Further, since the volume functional $|\cdot|$ is continuous (Schneider, 2014, Theorem 1.8.16), and given that $\mathscr{F}_c$ is the preimage of the closed set $[0,c]$, it follows that $\mathscr{F}_c$ is a closed subset of $\mathcal{A}$, and hence it is compact.

Observe next that maximizing $F(A) = \|\theta_X - \theta_Y\|_p^{(A)}$ is equivalent to maximizing $F^p(A)$, for $p > 0$, and as we show next $F^p(A)$ is upper semicontinuous under the Hausdorff metric, for every $A \in \mathscr{F}_c$. Let $A_n \to A$ in $(\mathscr{F}_c, d_H)$, for a fixed $c \ge 0$. It can be easily shown that (e.g., Schneider & Weil, 2008, Theorem 12.3.6)

$$1_A(t) \ge \limsup_n 1_{A_n}(t), \quad t \in T \subset \mathbf{R}^d. \tag{16}$$

Combining (16) with Fatou's lemma yields that

$$\begin{aligned}
F^p(A) &= \int_T 1_A(t)|\theta_X(t) - \theta_Y(t)|^p \, \mu(\mathrm{d}t) \\
&\ge \int_T \limsup_n 1_{A_n}(t)|\theta_X(t) - \theta_Y(t)|^p \, \mu(\mathrm{d}t) \\
&\ge \limsup_n \int_T 1_{A_n}(t)|\theta_X(t) - \theta_Y(t)|^p \, \mu(\mathrm{d}t) \\
&= \limsup_n F^p(A_n),
\end{aligned}$$

thus showing that $F^p(A)$ is upper semicontinuous under the Hausdorff metric, for every $A \in \mathscr{F}_c$. The final step of the proof is tantamount to a standard argument used for proving Weierstrass theorem. Let $u_c = \sup\{F^p(A) : A \in \mathscr{F}_c\} \cup \{\infty\}$. By definition, for every $c$ there exists a maximizing sequence $A_n \in \mathscr{F}_c$ such that $F^p(A_n) \to u_c$. By compactness, we can assume that $A_n \to A_c^*$. Upper semicontinuity of $F^p(A)$ implies that $u_c = \limsup_n F^p(A_n) \leq F^p(A_c^*)$, and on the other hand we have $u_c \geq F(A_c^*)$ since $u_c$ is the supremum. This proves that $u_c = F^p(A_c^*)$ is the maximum of $F^p(A)$, subject to $A \in \mathscr{F}_c$, and hence $A_c^*$ exists that solves (4).

**Claim 2**. First, note that

$$\|K\theta_X - K\theta_Y\|_p^{(A)} = \|\alpha + \beta\theta_X - (\alpha + \beta\theta_Y)\|_p^{(A)} = |\beta| \times \|\theta_X - \theta_Y\|_p^{(A)},$$

from where it follows that a set $A$ that maximizes $\|K\theta_X - K\theta_Y\|_p^{(A)}$ also maximizes $\|\theta_X - \theta_Y\|_p^{(A)}$; that is, $A_c^K = A_c^*$ for all $c \in [0, \infty)$. Second, it follows from the change of variables formula that

$$\|L\theta_X - L\theta_Y\|_p^{(A^L)} = \left[\int_{A^L}\{\theta_X(\alpha + \beta t) - \theta_Y(\alpha + \beta t)\}^p \mu(\mathrm{d}t)\right]^{1/p} = \frac{1}{\beta^{1/p}}\left[\int_A\{\theta_X(u) - \theta_X(u)\}^p \mu(\mathrm{d}u)\right]^{1/p},$$

where $A = \{\alpha + \beta t : t \in A^L\}$. Thus,

$$\arg\max\{\|L\theta_X - L\theta_Y\|_p^{(A^L)} : A^L \in \mathscr{F}_c\} = \arg\max\{\|\theta_X - \theta_Y\|_p^{(A)} : A \in \mathscr{F}_{\alpha+\beta c}\},$$

and hence, $A_c^L = A_{\alpha+\beta c}^*$.

**Claim 3**. Note first that $D_c^* \geq 0$, for all $0 < c < \infty$. Also, it holds that $D_c^* = 0$ if and only if $\theta_X = \theta_Y$ in $\Theta_c$; indeed, $D_c^* = 0$ implies that $0 = \|\theta_X - \theta_Y\|_p^{(A_c^*)} \geq \|\theta_X - \theta_Y\|_p^{(A)}$, for every $A \in \mathscr{F}^c$, which is only possible if $\theta_X = \theta_Y$ in $\Theta_c$, as $c > 0$. Finally, the triangle inequality for $\|\cdot\|_p^{(A)}$ and $\|F\|_\infty = \max_A |F(A)|$ yields that

$$\begin{aligned}
D_c^*(\theta_X, \theta_Z) &= \max\{\|\theta_X - \theta_Z\|_p^{(A)} : A \in \mathscr{F}_c\} \\
&= \max\{\|(\theta_X - \theta_Y) + (\theta_Y - \theta_Z)\|_p^{(A)} : A \in \mathscr{F}_c\} \\
&\leq \max\{\|\theta_X - \theta_Y\|_p^{(A)} + \|\theta_Y - \theta_Z)\|_p^{(A)} : A \in \mathscr{F}_c\} \\
&\leq \max\{\|\theta_X - \theta_Y\|_p^{(A)} : A \in \mathscr{F}_c\} + \max\{\|\theta_Y - \theta_Z)\|_p^{(A)} : A \in \mathscr{F}_c\} \\
&= D_c^*(\theta_X, \theta_Y) + D_c^*(\theta_Y, \theta_Z),
\end{aligned}$$

hence concluding the proof.

**Claim 4**. First, we note that an increase in $c$ represents augmenting the search domain as

$$\mathscr{F}_a \subset \mathscr{F}_b, \quad \text{for } b > a.$$

This combined with the fact that the set objective function $F(A) = \|\theta_X - \theta_Y\|_p^{(A)}$ is non-decreasing ($A \subseteq B$ implies $F(A) \leq F(B)$) yields that $D_b^* \geq D_a^*$, for $b > a$.

## B.2 Proof of Theorem 2

**Claim 1.** Tikhonov's theorem implies that from the Cartesian product of two compact sets results a compact set (Waldmann, 2014, Theorem 5.3.1). Hence, as a consequence of this, the search domain $T \times [0, R_c]$ is compact for every $c \geq 0$. Next, it is a routine exercise to prove that $f(t, r) = F\{B(t, r)\}$, is continuous for all $(t, r) \in T \times [0, R_c]$ as both $T$ and $[0, R_c]$ are compact. The final result then follows from Weierstrass theorem.

**Claim 2.** Let $\theta_X, \theta_Y \in L^p(T)$ be fixed and set $\mathscr{D}_c^* = \mathscr{D}_c^*(\theta_X, \theta_Y)$. The proof uses Berge's maximum theorem (Lemma 2) which, as can be seen from Appendix A, claims that a value function is continuous provided

that both the objective function and the constraint correspondence are continuous. In our setup, the value function is

$$\mathscr{D}_c^* = \max\{g(c,t,r) : (t,r) \in \varphi(c)\} = \max\{f(t,r) : (t,r) \in \varphi(c)\},$$

where $g(c,t,r) = f(t,r) = F\{B(t,r)\}$, and the constraint correspondence is $\varphi : [0,\infty) \twoheadrightarrow \mathbf{R}^{d+1}$, defined by

$$\varphi(c) = T \times [0, R_c], \qquad c \geq 0. \tag{17}$$

Thus, we just have to prove that $\varphi(c)$ in (17) is continuous, given that the objective function $g(c,t,r) = f(t,r) = F\{B(t,r)\}$ is trivially continuous for all $(t,r) \in T \times [0, R_c]$. Continuity of $\varphi(c)$ follows immediately from the product of correspondences theorem (Lemma 3), which yields that $\varphi(c) = \varphi_1(c) \times \varphi_2(c)$ is continuous as both $\varphi_1(c) = T$ and $\varphi_2(c) = [0, R_c]$ are compacted-valued, and $R_c$ is a continuous function for every $c \geq 0$. Finally, upper semicontinuity of the argmax correspondence,

$$\alpha_c = \{(t,r) \in T \times [0, R_c] : g(c,t,r) = \mathscr{D}_c^*\} = \{(t,r) \in T \times [0, R_c] : f(t,r) = \mathscr{D}_c^*\},$$

follows also from Berge's maximum theorem as $\mathbf{R}^{d+1}$ is Hausdorff.

### B.3   Proof of Theorem 3

Suppose by contradiction that $A'_{w,c}$ is the solution to the set function linear scalarization problem (14), for a fixed $w \in (0,\infty)^q$, but that $A'_{w,c}$ was not a Pareto optimal RMMD; then, there would exist a Pareto improvement $A \in \mathscr{F}_c$, with $A \neq A'_{w,c}$, so that $F_i(A) \geq F_i(A'_{w,c})$, for all $i$, and $F_i(A) > F_i(A'_{w,c})$, for at least an $i$. But then,

$$\sum_{i=1}^q w_i F_i(A) > \sum_{i=1}^q w_i F_i(A'_{w,c}),$$

which is a contradiction as $A'_{w,c}$ solves the set function linear scalarization problem.

## C   Monte Carlo Simulation Study

**Data generating processes and simulation settings**

We consider the following simulation scenarios based on Examples 1–2 from Section 2.1:

**Scenario 1 (Mean Functions)** BMDs between mean functions as in Example 1. Here, $X_t$ and $Y_t$ are Gaussian processes with mean functions given in (2) and where the same Matérn covariance function is assumed for both processes. Specifically $\mathrm{cov}(X_s, X_t) = \mathrm{cov}(Y_s, Y_t) = C_\nu(\|s - t\|_2, \sigma, \ell)$, where

$$C_\nu(d, \sigma, \ell) = \sigma^2 \frac{2^{1-\nu}}{\Gamma(\nu)} \left(\sqrt{2\nu} \frac{d}{\ell}\right)^\nu K_\nu\left(\sqrt{2\nu} \frac{d}{\ell}\right),$$

for $(s,t) \in [0,1]^2$, where $K_\nu$ is the modified Bessel function (Abramowitz & Stegun, 1964, Section 9.6), and where $\sigma, \nu$, and $\ell$ are positive parameters, here set as $\sigma = \nu = \ell = 1$. The simulated data are then a discretized version of $n$ simulated Gaussian processes evaluated over a grid on the unit interval,

$$\mathcal{D}_X = \{X_{t,i} : t \in \{0/J, \ldots, (J-1)/J\}\}_{i=1}^n,$$

with $n \in \{10, 50, 100, 200\}$ and $J \in \{10, 20\}$; the same comments apply to $\mathcal{D}_Y$.

**Scenario 2 (Intensity Functions)** BMDs between intensity functions as in Example 2. Here, points drawn from non-homogeneous bivariate Poisson process with mean measures,

$$\begin{cases} E\{N_X(A)\} = \int_A \gamma \exp\{-(t_1^2 + t_2^2)/2\} \, \mathrm{d}t, \\ E\{N_Y(A)\} = \delta E\{N_X(A)\}, \end{cases} \tag{18}$$

for $A \subseteq T = [-3,3]^2$. While the sample sizes in this scenario are random quantities, given by $N_X = N_X(T)$ and $N_Y = N_Y(T)$, the mean number of simulated points over $T$ is $(E(N_X), E(N_Y)) \approx (2\pi\gamma, 2\pi\gamma\delta)$, a simple yet accurate approximation that follows immediately from multiple Gaussian integrals. The simulated data are given by the following collection of points

$$\mathcal{D}_X = \{(X_{1,1}, X_{2,1}), \dots, (X_{1,N_X}, X_{2,N_X})\},$$

with $\gamma \in \{25, 50\}$ and $\delta \in \{2, 4, 8, 16\}$, and where $\mathcal{D}_Y$ is analogously defined.

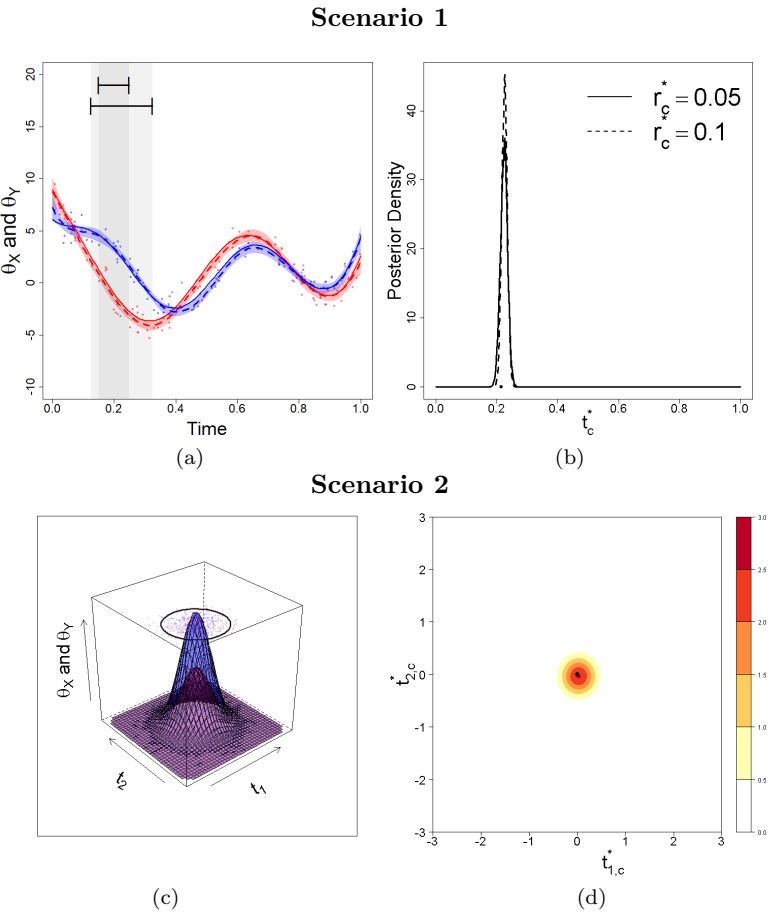

Figure 5: One shot experiments for Scenarios 1 and 2 (a) Black solid segments corresponds to fitted BMDs based on the simulated Gaussian process data; the chart also depicts the fitted functional parameters (dashed) along with 95% credible bands, and the true parameters (solid). (b) Marginal posterior density for the center $t_c^*$ plotted against the true center (rug). (c) and (d) are identical to (a) and (b), respectively, but for the simulated point process data from Scenario 2.

**Prior specification and posterior inference**

Inferences for the BMDs were carried out by sampling $m = 1\,000$ and $500$ times using Algorithm 1 for Scenarios 1 and 2, respectively. As can be seen from Algorithm 1, inferences for BMDs are constructed from the functional parameters, and thus we now comment on what versions of (8) have been used for fitting the latter. For Scenario 1 the identity link function was used in (8) along with B-spline basis, and the number of basis functions was selected using the DIC. The default uninformative priors of `R-INLA` have been used, which consist of diffuse priors for the $\beta$'s—i.e. $\beta_0 \sim N(0, \infty)$ and $\beta_i \sim N(0, 1000)$—and a long-tailed prior for the variance of the error term—i.e. a log gamma distribution, where the gamma distribution has mean $a/b$ and variance $a/b^2$, with $a = 1$ and $b = 10^{-5}$; see Wang et al. (2018, Section 5.2.1) for further

details. For Scenario 2 we follow Simpson et al. (2016) and specify a log-Gaussian Cox process using (8) by setting a log link function, that links the intensity function with a Matérn random field using piecewise linear basis functions over a mesh, and where the $\beta$'s are Gaussian-distributed. For the parameters of the Matérn covariance function we use the PC prior approach of Fuglstad et al. (2019) setting $P(\sigma > 1) = 0.001$ and $P(\ell < 0.05) = 0.001$.

**One shot experiments**

We first illustrate the methods on a single run experiment for some instances of Scenarios 1 and 2. In Fig. 5(a) we show the fitted BMDs for Scenario 1, along with the corresponding mean functions, on a one shot experiment with $n = 10$ and $J = 10$, for $c = 0.1$ and $c = 0.2$. As can can be seen from Fig. 5(a), the fitted BMDs accurately recover the true $B_{0.1}^* = [0.165, 0.265]$ and $B_{0.2}^* = [0.115, 0.315]$. In Fig. 5(b) we also display the marginal posterior density for the optimal center $t_c^*$ which quantifies the uncertainty surrounding the true. The marginal posterior for the radius is essentially degenerated, as predicted earlier in the comments surrounding (7), and hence not shown.

In Fig. 5(c) we depict the fitted BMD for Scenario 2, on a one shot experiment with $\gamma = 25$ and $\delta = 2$, for $c = 2\pi$, and display the fitted intensity functions. As it is evident from Fig. 5(c), the fitted BMD nicely uncovers the true, and indeed it completely overlaps it to the point that the true (depicted in gray) is barely visible.

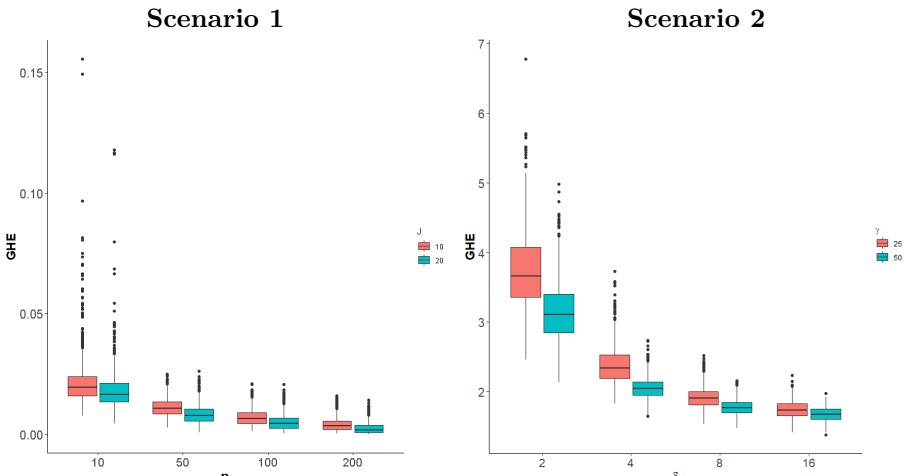

Figure 6: Side-by-side boxplots of GHE in (19) for Monte Carlo simulation study.

**Monte Carlo evidence**

We redo the previous one shot analyses $M = 1\,000$ times, considering different samples sizes, and relying on the GHE (Posterior Mean **G**lobal **H**ausdorff **E**rror),

$$\text{GHE} = E\left\{ \int_0^{|T|} D_{\text{H}}(B_c^*, \widehat{B}_c^*) \, \mathrm{d}c \, \middle| \, \mathcal{D}_X, \mathcal{D}_Y \right\} \tag{19}$$

to quantify how accurate on average are the estimated BMDs, $\widehat{B}_c^*$, over $0 \le c \le |T|$. In Scenario 1 we use $D_{\text{H}}(B_c^*, \widehat{B}_c^*) = \max\{|(t_{0,c}^* - r_c^*) - (\widehat{t}_{0,c}^* - \widehat{r}_c^*)|, |(t_{0,c}^* + r_c^*) - (\widehat{t}_{0,c}^* + \widehat{r}_c^*)|\}$, while in Scenario 2 we use a numerical approximation of $D_{\text{H}}(B_c^*, \widehat{B}_c^*)$ implemented using Borchers (2021). Finally, the GHE for each simulated dataset is computed as

$$\text{GHE}_j = \frac{1}{m} \sum_{i=1}^{m} \int^{|T|} D_{\text{H}}(B_c^*, \widehat{B}_c^{*[i,j]}) \, \mathrm{d}c,$$

where $\widehat{B}_c^{*[i,j]}$ is the $i$th posterior sampled BMD, based on the $j$th simulated sample, for $i = 1, \ldots, m$ and $j = 1, \ldots, M$. As can be seen from Fig. 6, GHE tends to decrease as $n, J, \gamma$, and $\delta$ increases. Such behavior confirms the expected frequentist behavior of the methods, as $n$ and $J$ dictates the amount of simulated data for Scenario 1, and $\gamma$ and $\delta$ does the same for Scenario 2. To put it differently, since larger values of $\gamma$ and $\delta$ imply larger sample sizes, the observed reduction in GHE as a function of the latter parameters suggests a sensible asymptotic performance of the proposed Bayesian inferences.

# D  Selected Comments on the Choice of $c$

## D.1  The Coincidence Set Criterion

Optimal data-driven selection of $c$ requires a criterion or loss function and it remains an open problem for future analysis. Motivated by a reviewer's comment, this section offers hints on potential starting points.

For $T = [a, b] \subset \mathbb{R}$, a practical approach is to center the parameter functions and establish a criterion based on the width of intervals defined by their coincidence point(s); we term this the coincidence set criterion and will detail it next. While parts of the construction can be readily extended beyond the real line, fully developing the criterion in $\mathbb{R}^d$ are outside the scope of this section.

Formally, define the demeaned parameter functions as

$$\tilde{\theta}_X(t) = \theta_X(t) - \frac{1}{b-a}\int_a^b \theta_X(u)\,\mathrm{d}u, \qquad \tilde{\theta}_Y(t) = \theta_Y(t) - \frac{1}{b-a}\int_a^b \theta_Y(u)\,\mathrm{d}u, \tag{20}$$

and the coincidence set as

$$\Pi = \{t : \tilde{\theta}_X(t) = \tilde{\theta}_Y(t)\}. \tag{21}$$

In words, $\Pi$ is the set of points where $\tilde{\theta}_X$ and $\tilde{\theta}_Y$ coincide.

**Proposition 1.** *Assume $\theta_X$ and $\theta_Y$ are continuous functions on $T = [a,b]$ and differentiable on $(a,b)$. Then, the coincidence set is nonempty, that is, $\Pi \neq \emptyset$.*

*Proof.* The proof follows from the mean value theorem (e.g., Tao, 2006, Corollary 10.2.9). First, note that

$$\Pi = \{t : \tilde{\theta}_X(t) - \tilde{\theta}_Y(t) = 0\}$$
$$= \left\{t : \theta_X(t) - \theta_Y(t) = \frac{1}{b-a}\int_a^b \theta_X(u) - \theta_Y(u)\,\mathrm{d}u = 0\right\}$$
$$= \left\{t : \delta(t) = \frac{\Delta(b) - \Delta(a)}{b-a}\right\},$$

where $\delta(t) = \theta_X(t) - \theta_Y(t)$ and $\Delta(b) - \Delta(a) = \int_a^b \delta(u)\,\mathrm{d}u$. The mean value theorem implies that there exists an $c \in (a,b)$ such that

$$\delta(c) = \frac{\Delta(b) - \Delta(a)}{b-a},$$

from where the final result follows.

$\square$

To streamline the presentation of ideas, we first consider the case where $\tilde{\theta}_X(t)$ and $\tilde{\theta}_Y(t)$ touch at only a finite number of points, i.e., where $\Pi$ has Lebesgue measure 0. In that case, the criterion can be written as

$$c_{\mathrm{sc}} = \max\{\tau_{i+1} - \tau_i\}_{i=0}^m, \tag{22}$$

where $m < \infty$ is the cardinality of $\Pi$ and $a = \tau_0 < \tau_1 < \cdots < \tau_m < \tau_{m+1} = b$, with $\tau_1, \ldots, \tau_m$ denoting the ordered elements of $\Pi$.

The extension of (22) to the general case—that takes into account parameter functions could coincide over a set $\Pi$ with positive measure and at an infinite number of points—is similar but slightly more technical. First, it is convenient to partition the coincidence set as follows

$$\Pi = \Pi_0 \cup \Pi_1,$$

where $\Pi_0 = \{\tau_i\}_{i=1}^M$ is a discrete set (with the same interpretation as above, but with $M \leq \infty$) and $\Pi_1 = \bigcup_{j=1}^J (\alpha_j, \beta_j)$ is a collection of $J \leq \infty$ open sets. The general definition of the coincidence criterion is

$$c_{\text{sc}} = \max \left\{ \tau_1 - a, \sup\{\tau_{i+1} - \tau_i\}_{i=1}^{M-1}, b - \lim_{i \to M} \tau_i \right\}, \tag{23}$$

where $\{\tau_i\}_{i=1}^M$ are the ordered elements of $\Pi_0$ and $M \leq \infty$. Note that if $\tilde{\theta}_X(t)$ and $\tilde{\theta}_Y(t)$ touch at only a finite number of points, then $\Pi = \Pi_0$ and hence (23) becomes (22) with $M = m$.

## D.2   Data Illustrations

Fig. 7 illustrates the BMDs obtained using this experimental criterion with two real data examples. The findings are comparable to those presented in the main paper, although the regions are slightly larger.

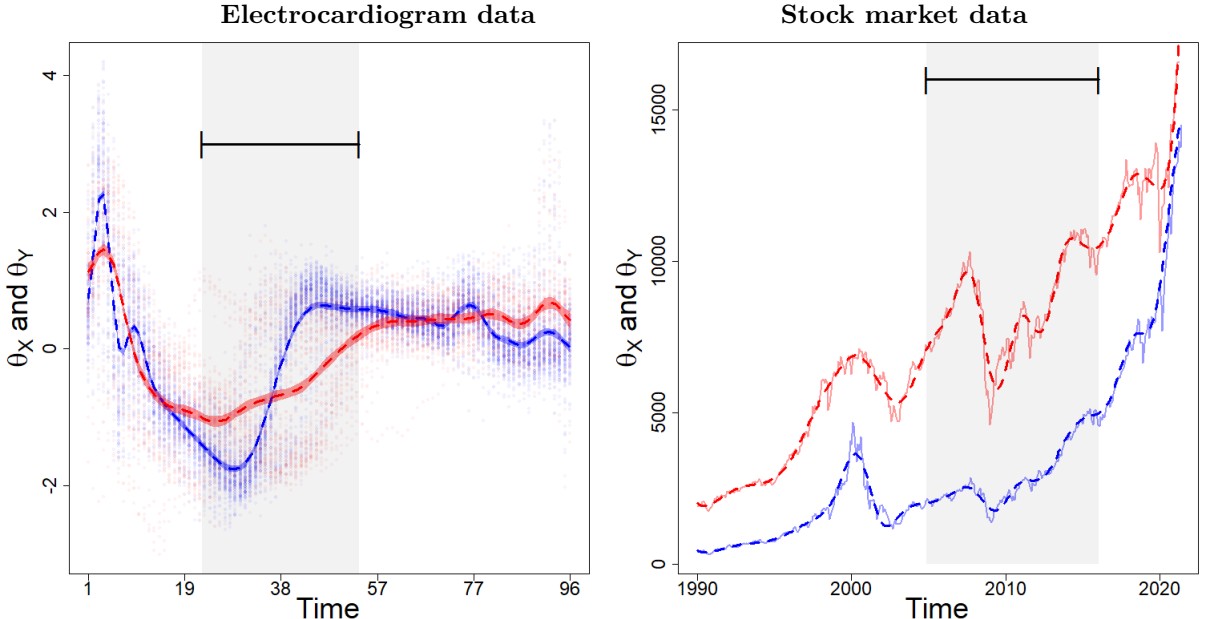

Figure 7: BMDs corresponding to the data driven $c$ obtained via the coincidence set approach.

## E   List of Symbols

Table 1 lists symbols and notation used throughout the article.

| Symbol | Description |
|--------|-------------|
| $X, Y$ | stochastic processes under comparison, that is, $X = \{X_t\}$ and $Y = \{Y_t\}$ |
| $T$ | ground, or index, set over which processes $X$ and $Y$ are defined |
| $\theta_X, \theta_Y$ | functional parameters for $X$ and $Y$ |
| $\mathcal{D}_X, \mathcal{D}_Y$ | data on proceses $X$ and $Y$ |
| $\|\cdot\|_p^{(A)}$ | $L^p$ sub-norm over $A$ |
| $F(\cdot)$ | set objective function $\|\theta_X - \theta_Y\|_p^{(\cdot)}$ |
| $f(t, r)$ | set objective function evaluated at closed ball $B(t, r)$, that is, $F\{B(t, r)\}$ |
| $F_i$ | set objective functions in multi-objective context |
| $\mathcal{A}$ | set of compact and convex subsets of the ground set $T$ |
| $\|A\|$ | volume functional, i.e., Lebesgue measure of $A$ in $\mathbf{R}^d$ |
| $\mathscr{F}_c$ | collection of feasible sets |
| $A_c^*$ | region of maximum or multi-maximum dissimilarity |
| $D_c^*$ | dissimilarity index for region of maximum dissimilarity |
| $B_c^*$ | ball of maximum or multi-maximum dissimilarity |
| $\mathscr{D}_c^*$ | dissimilarity index for ball of maximum dissimilarity |
| $t_c^*, r_c^*$ | center and radius of ball of maximum dissimilarity |
| $\mathscr{B}_p$ | set of all closed balls in $L^p$ |
| $d_\mathrm{H}(A, B)$ | Hausdorf distance between sets $A$ and $B$ |
| $d(x, B)$ | minimum Euclidean distance between a point $x \in A$ and set $B$ |
| $\partial A$ | boundary of set $A$ |

Table 1: Symbols and notation used throughout the article.

