# OpenReview forum: "Uncovering Sets of Maximum Dissimilarity on Random Process Data"
_TMLR — Accepted by TMLR_

### Review · Reviewer_kWQx · 2024-04-02

**Summary Of Contributions:**

The authors introduce a new problem formulation of discovering the region where two stochastic processes differ the most. They then map it to a canonical problem in set function optimization and prove that a solution exists. For a special case where the regions are defined as closed balls, the authors propose a practical Bayesian algorithm for finding the solution, building on INLA.

The otherwise highly theoretical paper is grounded in reality by three very clear empirical experiments that both make the concept clear and demonstrate the computational method works well.

**Audience:**

Yes

**Broader Impact Concerns:**

No concerns.

**Claims And Evidence:**

Yes

**Requested Changes:**

Some recommendations on improving the writing were made when listing the weaknesses, but other than that I do not have any requests.

One visual aspect that you could think about: In Figure 2 (a) the set is communicated very nicely as visually apparent circle with thick borders, but for all of the univariate cases (Fig 1 (a), (c); Fig 3; Fig 4) the intervals are a bit difficult to see. This was particularly problematic in Fig 1 that uses gray color instead of black and I only noticed the invervals at a rather late stage. You should at least make the intervals black in Fig 1, but more generally could consider making the intervals more visible. Even though the intervals relates to the x-axis, is does not need to be visually drawn only on top of the axis.

**Strengths And Weaknesses:**

Overall, the paper is an elegant combination of rather involved theory and highly practical demonstrations. I personally do not have sufficient background for ensuring the correctness of the theoretical properties, but the problem formulation itself is novel and clearly useful and the empirical demonstrations show the method works. These are alone sufficient for a good paper.

Main strengths:
- The problem formulation itself is clear and Figure 1 supports understanding the setup early on, even if not following the mathematical basis in Introduction well.
- The empirical experiments are extremely nice, showcasing the diversity of the possible applications this could be used for. They are simple enough so that everyone can understand the point, but still real examples where a practitioner could actually be interested in the result and was missing practical tools before this. I can see many spatial statisticians getting interested in Section 4.1 and there are also other signal processing applications like Section 4.3 where this method could be used e.g. for feature selection (as already hinted by the authors).
- The work is in general exemplary in how it combines theory and practice, covering the whole range from mapping an interesting problem formulation to abstract mathematical concepts and then making it very concrete in terms of an easy-to-follow algorithm and clear empirical results. I will most likely be using this as educational example in some occasions.
- Using Bayesian inference (rather than just finding the most likely ball -- I presume you could do that just as well?) is a nice additional touch which further increases the potential impact.

Minor weaknesses:
- The notion of set functions in Introduction remains a bit too vague. Maybe the Introduction would work better if you first discussed the practical examples a bit, explaining the concept of the region a bit more formally etc, and only then made the connection to set functions? Then it would be easier to make concrete what $A$, $\mathcal{A}$ etc are in practice in some example optimization tasks? I still struggle to make the connection between the regions and the set functions clear even after reading through the whole paper.
- Section 2.2 is a bit condensed. I believe a lot of readers would benefit if the information around Eq (7) was expanded. You now state fairly fundamental properties ("can actually be written as a standard continuous optimization problem") but do not really show it. I believe this is not a novel result of your work, so simply referring to other papers that discuss this equivalence in more detail would already help, and you could also spend a few more lines providing the intuition and importance for readers who are not so familiar with the connection. You could also motivate a bit more on why you choose to consider the posterior instead of simply solving (7).

---

> ### Author Response · Authors · 2024-05-06
> **Detailed Response to Reviewer kWQx**
>
> Dear Reviewer kWQx,
>
> Thank you for the detailed feedback and the thoughtful summary of our work. Please find our revisions detailed below along with some clarifications motivated by your valuable input.
>
>
> ### Minor Weaknesses & Requested Changes
>
> #### Notion of set functions:
> - Based on your feedback, we have introduced a gentler start to set functions in the Introduction, making it more concrete with examples. We note that, probability measures, F(A) = Pr(A), and the counting measure F(A) = #A are examples of set functions.
>
> - We now clarify from the outset that, loosely speaking, any function mapping a set A to a real value can be understood as a set function.
>
> #### Eq. (7):
> - Motivated by your feedback, we have expanded the discussion on computing and optimization. We also added a sentence before the section on balls of maximum dissimilarity to highlight their computational convenience as they parameterize RMDs.
>
> - We hope these changes emphasize that balls of maximum dissimilarity, parameterized by the center-radius pair, are suitable approximations to RMDs. Thus, finding an optimal ball corresponds to a standard constrained optimization problem, where the objective function depends solely on the center-radius vector with $d+1$ components. Thank you again.
>
> - Finally, we added some brief comments surrounding Algorithm 1 noting that we do solve (7) per each INLA iteration as noted in Step 3 of Algorithm 1 and the posterior quantifies the uncertainty surrounding the optimal centre and radius.
>
> #### Aesthetics and visual aspects:
> - We have carefully revised the visualizations hoping they are now more easily understood and visible.
>
> Thanks to your guidance, the paper has greatly benefited—thank you once again for your valuable input and support.

---

### Review · Reviewer_epMy · 2024-04-05

**Summary Of Contributions:**

This paper considers analyzing random process data (like time series, functional
data, and point processes) to identify regions where two processes differ the
most statistically. The authors formulate the problem of finding the regions of
maximum dissimilarity and balls of maximum dissimilarity as a continuous set
function optimization problem with a volume constraint. Theoretical properties
of RMD and BMD are studied and multiple real data exams are provided.

**Audience:**

Yes

**Broader Impact Concerns:**

I don't see any concerns on the ethical implications.

**Claims And Evidence:**

Yes

**Requested Changes:**

Requested Changes*

1. Provide more details on the notations.
2. Make the figures more readable on gray scale. Some figures such as figures
   1(a), 1(c), and figure 3 are difficult to read when printed on papers.
3. Discuss the challenges and/or differences for working with continuous set
   optimizations as compared with discrete sets.
4. Give some insights on the connections between regions and balls.

**Strengths And Weaknesses:**

Strengths:

1. The paper introduces a novel concept of sets of maximum dissimilarity on
   random process data, which can be applied to various types of data, including
   time series, functional data, and point processes.
2. The proposed method is formulated as a continuous set function optimization
   while existing work in this area has focused on discrete or combinatorial
   sets.

Weaknesses:

1. The paper lacks a detailed discussion of the computational complexity of the
   proposed method, especially for large-scale data sets. Usually, a continuous
   optimization problem is much easier to solve than a discrete one, as the
   latter is often NP-hard.
2. Some notations and expressions needs further details. For example, below (1),
   what do you mean by 2 to the power of a set? What are the differences and
   connections between regions of maximum dissimilarity and balls of maximum
   dissimilarity? Why only consider learning about balls of maximum
   dissimilarity from Data?

---

> ### Author Response · Authors · 2024-05-06
> **Detailed Response to Reviewer epMy**
>
> Dear Reviewer epMy,
>
> Thank you for the detailed feedback and the thoughtful summary of our work. Please find our revisions detailed below along with some clarifications motivated by your valuable input.
>
> ### Weaknesses & Requested Changes
>
> #### Notations:
> - We have revised parts of the text to enhance notation details
>
> - Added a list of symbols and notation to the appendix.
>
> #### Figures:
> - We have revised and enhanced the figures and hope they are more readable now.
>
> #### Continuous vs Discrete:
> - Thank you for another insightful comment. Based on your feedback, we now emphasize from the outset that the learning problem discussed in our article inherently involves continuous set function optimization, as it seeks to find a subset of the continuum with positive measure (loosely, a ‘continuous set’) where differences are maximized.
>
> - The complexity of the search domain—a set of ‘continuous sets’—makes continuous set function optimization significantly more sophisticated than its discrete counterpart. Establishing even the existence of a solution is nontrivial, as demonstrated by Theorem 1 (claim 1). As noted in the paper, our work, prompted by the unique challenges of the learning problem we introduce in §1, represents an initial step towards advancing continuous set function optimization.
>
> For practical considerations, entailed in the estimation of RMDs from data, we introduced the balls of maximum dissimilarity as a suitable parametric approximation to RMDs.
>
> - Discrete analogues of our problem may also be of interest in their own right. Therefore, based on your feedback, we added a note on the potential of using discrete approaches to provide alternative ways of approximating continuous RMDs.
>
> #### Connections between regions and balls
> - Motivated by your comment, we have updated §2.1 to clarify that balls of maximum dissimilarity serve as a suitable parametric approximation to RMDs, thereby facilitating computations. To enhance this connection, we included two sentences on page 5, along with more details surrounding (7), and hope their relationship is now clearer.
>
> Thanks to your guidance, the paper has greatly benefited—thank you once again for your valuable input and support.

---

> > ### Comment · Reviewer_epMy · 2024-05-20
> > **Paper improved**
> >
> > Thank you for addressing my comments, especially on the point of Continuous vs Discrete. Now I am more clear about the computational challenge of the problem. Just a minor point: change the period in (7) to a comma and remove "then" in "then the optimal radius ..." two lines below (7).

---

> > > ### Author Response · Authors · 2024-05-21
> > > **Done**
> > >
> > > Thank you for your reply and feedback—much appreciated. We have incorporated these final comments in the manuscript.

---

### Review · Reviewer_BR9h · 2024-04-27

**Summary Of Contributions:**

This paper proposes a new dissimilarity measure to identify a region where two stochastic processes differs most. The paper is well written, the proposed idea is interesting and can be useful in practice.

**Audience:**

Yes

**Broader Impact Concerns:**

No concerns

**Claims And Evidence:**

Yes

**Requested Changes:**

In my opinion, this paper essentially provides a new graphical visualization tool to identify a region where two stochastic processes are most different, similar to a box plot or a functional box plot. I can imagine that such a tool can be useful for many researchers who work with stochastic processes. Therefore, my overall opinion on the proposal is positive. The only concern I have is that the volume of the set (i.e., $c$) needs to be given in advance, which is unrealistic in practice. A practical guideline on how $c$ should be chosen in practice should be provided. I am curious about the following questions.

1. One can imagine that even if two processes are generated from the same model, one can still identify such a region with maximal dissimilarity, and how would such a region be different from two processes that have very different data-generating mechanisms?

2. If there is no difference between two stochastic processes, what would be the distribution of calculated dissimilarity measures? (with a given $c$ or as a function of $c$?)

**Strengths And Weaknesses:**

Strengths: the paper is well written, the presentation is clear, the idea is interesting.
Weakness: the choice of $c$, the volume of the dissimilarity set needs to be given, which is difficult in practice.

---

> ### Author Response · Authors · 2024-05-06
> **Detailed Response to Reviewer BR9h**
>
> Dear Reviewer BR9h,
>
> Thank you for the detailed feedback and the thoughtful summary of our work. Please find our revisions detailed below along with some clarifications motivated by your valuable input.
>
> ### Requested Changes
>
> #### Choice of “c”:
> - Following your comment, we have added a disclaimer when opening §2.1 to guide readers that in the context of this paper, $c$ should be set by the user based on the desired level of granularity; this enables tracking of maximum differences on a weekly, monthly, or annual basis, for example. (Say, looking for differences at six months, one year, and two years correspond to $c = 6,12,$ and 24, respectively, as illustrated in the data application in §4.2.)
>
> - Based on your feedback, we have added a sentence to the Discussion section, openly acknowledging that—while the paper focuses on scenarios where the user sets $c$—it would be natural to integrate the choice of $c$ into a decision theory framework. While this remains an open problem, motivated by your suggestions, Appendix E now provides hints on starting points for future analyses.
>
> #### Questions:
>
> - Q2. Distribution of $\mathscr{D}_c^*$ when $X = Y$: Thank you for another insightful comment. Our numerical experience with the method suggests that the posterior distribution of the dissimilarity, $\mathscr{D}_c^*$ concentrates around zero on that case, which aligns with expectations. Although we lack formal proof of consistency—the evidence presented in Fig. 6 supports this hypothesis.
>
> - Q1: $X = Y$ versus $X \neq Y$: Maximum dissimilarity regions have been conceived as a tool for highlighting where a potential difference may exist. In the important yet pathological case where $X = Y$, spurious regions could be observed when fitting BMDs in practice. Yet, as noted above, our experience with method leads us to conjecture that, the posterior distribution of $\mathscr{D}_c^* > 0$ as $n \to \infty$, when if $X \neq Y$, whereas $\mathscr{D}_c^* \to 0$ if $X \neq Y$, as $n \to \infty$. This could be perhaps useful for assessing the statistical significance of an obtained set of maximum dissimilarity, for the case $X = Y$. Yet, as this is the pioneering paper on sets of maximum dissimilarity, we currently lack a formal asymptotic analysis and on top of this, $\mathscr{D}_c^* = 0$ is a boundary problem.
>
> We added one sentence to the Discussion to openly acknowledge this as an open problem for future analysis.
>
> Thanks to your guidance, the paper has greatly benefited—thank you once again for your valuable input and support.

---

### Comment · Action_Editor_wsED · 2024-05-27
**Please add the URL link to your camera ready**

Currently the `id` in the URL is just `XXXX`

---

> ### Author Response · Authors · 2024-05-28
> **Done**
>
> We have fixed this and hope that now it is okay. Thank you.

---

> > ### Comment · Action_Editor_wsED · 2024-05-28
> > **Still not right**
> >
> > Now the link points to a particular comment -- please fix the link to point to the review forum
> >
> > ```
> > https://openreview.net/forum?id=ntWCJrlDD8
> > ```

---

### Decision · Action_Editor_wsED · 2024-05-20

**Recommendation:** Accept as is

**Comment:**

The reviewers generally agreed that the paper is a solid piece of work with no obvious limitations. Claims are reasonable and supported by evidence, and the resulting applications in visualizing discrepancies in stochastic processes will be of interest to the community.

**Audience:**

There is certainly a subset of the TMLR audience that would find the application of this method---i.e., for augmenting visualizations of stochastic processes to highlight discrepancies---of interest.

**Claims And Evidence:**

The paper proposes a method for finding regions of discrepancy between stochastic processes. The paper formulates a set optimization problem, and studies some high-level theoretical properties of the problem (existence, behaviour under transformations, etc). It then proposes a restriction of the problem to ball-shaped sets, studies theoretical properties of that problem, develops a method to approximately solve the optimization problem, and applies it to a few illustrative examples. Theoretical claims are supported with proofs, and claims that the method works are justified reasonably by the empirical results.

---

> ### Author Response · Authors · 2024-05-22
> **Camera-Ready Version**
>
> Dear Action Editor,
>
> The camera-ready version of our paper has been uploaded. Please let us know if anything else is required. We appreciate the insightful feedback from the reviewers and yourself which has greatly enhanced our work.
>
> Best wishes,
>
> The Authors